# Propensity for somatic expansion increases over the course of life in Huntington disease

Radhia Kacher[1,2], François-Xavier Lejeune[1,3], Sandrine Noël[4], Cécile Cazeneuve[4], Alexis Brice[1], Sandrine Humbert[2]*, Alexandra Durr[1,4]*

[1]Sorbonne Université, Paris Brain Institute (ICM Institut du Cerveau), AP-HP, INSERM, CNRS, University Hospital Pitié-Salpêtrière, Paris, France; [2]Univ. Grenoble Alpes, INSERM, U 1216, Grenoble Institut Neurosciences, Grenoble, France; [3]Paris Brain Institute's Data and Analysis Core, University Hospital Pitié-Salpêtrière, Paris, France; [4]Neurogenetics Laboratory, Department of Genetics, Assistance Publique–Hôpitaux de Paris, University Hospital Pitié-Salpêtrière, Paris, France

**Abstract** Recent work on Huntington disease (HD) suggests that somatic instability of CAG repeat tracts, which can expand into the hundreds in neurons, explains clinical outcomes better than the length of the inherited allele. Here, we measured somatic expansion in blood samples collected from the same 50 HD mutation carriers over a twenty-year period, along with post-mortem tissue from 15 adults and 7 fetal mutation carriers, to examine somatic expansions at different stages of life. Post-mortem brains, as previously reported, had the greatest expansions, but fetal cortex had virtually none. Somatic instability in blood increased with age, despite blood cells being short-lived compared to neurons, and was driven mostly by CAG repeat length, then by age at sampling and by interaction between these two variables. Expansion rates were higher in symptomatic subjects. These data lend support to a previously proposed computational model of somatic instability-driven disease.

*For correspondence:
sandrine.humbert@univ-grenoble-alpes.fr (SH);
alexandra.durr@icm-institute.org (AD)

## Introduction

Most mutations are stably transmitted from parent to offspring. This reliable genetic principle does not hold, however, for dynamic mutation disorders such as Fragile X syndrome or Huntington disease (HD). In these diseases, a sequence such as a CAG repeat tract can expand during transmission, likely through mechanisms involving replication or transcription (*Khristich and Mirkin, 2020*). In general, the longer the repeat, the earlier the patient develops overt symptoms and the more aggressive the disease is likely to be (*Koshy and Zoghbi, 1997*). Thus, in HD, modest expansions of 40 repeats in huntingtin gene (*HTT*) are associated with the appearance of motor, cognitive, and psychiatric disturbances in mid- or late adulthood, whereas large expansions of over 80 repeats cause childhood onset with additional features such as epilepsy and a more rapidly fatal course (*Bates et al., 2015b*; *Sun et al., 2017*). Yet on an individual subject basis, we cannot predict the disease course just from the size of the repeat tract: two individuals with the same length repeat expansion in *HTT* may experience disease onset decades apart (*Andrew et al., 1993*). The inherited pathological CAG repeat size accounts for about 42–71% of the age at onset in HD (*Squitieri et al., 2006*), though the confidence limits narrow for tracts longer than 50 CAGs (*Andrew et al., 1993*; *Bates et al., 2015b*; *Langbehn et al., 2004*; *Rubinsztein et al., 1997*; *Wexler et al., 2004*).

Part of the reason for such variability could be that HD is still thought of primarily as a movement disorder, so age at onset in HD is typically defined as the point at which motor symptoms become unequivocal. But abnormalities in the brain are present from early development (*Barnat et al.,*

*2020*), and mutation carriers may experience cognitive deficits, psychiatric disturbances, or even subtle motor impairments years before diagnosis (*Bates et al., 2015b*). It is challenging, and somewhat misleading, to pinpoint age at onset in a disease that evolves insidiously like HD.

A more interesting explanation takes into account the fact that CAG repeats do not just expand in the germline. They are also somatically unstable, such that different CAG expansions can be identified in the same sample tissue from various organs and brain regions. Somatic mosaicism occurs both in mouse models of HD (*Kennedy and Shelbourne, 2000*; *Larson et al., 2015*; *Lee et al., 2010*) and in humans with HD (*Kennedy et al., 2003*; *Swami et al., 2009*; *Telenius et al., 1994*). The greatest increases in CAG tract length have been observed in the brain regions most affected in HD, the cerebral cortex and the striatum, whose neurons can harbor repeat tract expansions in the hundreds (*Gonitel et al., 2008*; *Kennedy et al., 2003*; *Møllersen et al., 2010*; *Mouro Pinto et al., 2020*; *Shelbourne et al., 2007*). Repeat expansions likely result from the formation of unusual DNA structures that predispose the tract to errors in mismatch repair (*Khristich and Mirkin, 2020*; *Tabrizi et al., 2020*). In fact, variants in several different DNA repair genes are associated with somatic instability in both animal models of HD (*Dragileva et al., 2009*; *Pinto et al., 2013*; *Tomé et al., 2013*) and HD patients (*Ciosi et al., 2019*; *Flower et al., 2019*; *Lee et al., 2019*).

Mounting evidence suggests that somatic repeat lengths better explain age at onset than the germline repeat, as their propensity to expand relates to both the baseline allele length and age (*Ciosi et al., 2019*; *Lee et al., 2019*). Interestingly, these data lend support to a mathematical model put forth over a decade ago (*Kaplan et al., 2007*). In brief, Kaplan et al. proposed that the onset and progression of triplet repeat diseases, including HD, are determined by the rate of somatic expansion in disease-relevant cells. Symptoms manifest when a critical proportion of cells (say, 20%) pass a pathogenic threshold, which would differ for different cell types. Their modeling suggests the threshold in striatal neurons for HD would be ~115 repeats. They further posited that at birth, nearly all cells would carry just the inherited number of repeats, but that over time the mutant alleles would further expand at a rate that increased linearly with the number of repeats. The rate of expansion would thus determine how rapidly the pathological state is reached, and thus should influence disease onset and progression.

The Kaplan model is quite compelling, but to test its predictions requires longitudinal data to study the evolution of somatic instability over time within patients. Given that the *HTT* mutation was discovered less than thirty years ago, such a study is only now becoming feasible. Even so, there are limits to how much of the model can be tested in humans. We cannot, for example, sample neurons over the life span to see how many come to exceed 115 repeats, or tally the proportion of neurons that reach a pathogenic threshold before phenoconversion. Nevertheless, we have been able to measure somatic repeat expansions in blood samples from HD carriers and patients over a twenty-year period and examine cortical tissue from mutation-carrying fetuses and deceased adults. By characterizing the degree of somatic expansion at these different stages, we were able to analyze associations between changes in the somatic expansion, age, and inherited CAG repeat length.

## Results

### Determination of somatic expansion index in HD carriers

We collected biological samples from 72 HD mutation carriers across the life span: 7 fetuses, 50 adults, and 15 post-mortem brains (see *Tables 1*, *2* and *3*). For all samples, we calculated an expansion index (EI) based on a specific PCR followed by fragment sizing to identify the peaks corresponding to different numbers of CAG repeats, or (CAG)n (*Lee et al., 2010*; *Mouro Pinto et al., 2020*). The expanded allele has a characteristic PCR profile with one particularly prominent peak, which provides the CAG repeat size given for diagnosis (*Figure 1—figure supplement 1A*, see 'Materials and methods'). This 'reference peak' is flanked by additional peaks that reveal the various repeat lengths in a given tissue, which we refer to as mosaicism or somatic instability. The fluorescence intensity of each peak reflects the proportion of cells bearing each somatic expansion, but it is worth noting that PCR is biased toward alleles containing smaller repeats. Because peaks to the left of the reference peak can be generated by polymerase slippage during PCR, we used only those to the right of the main peak to calculate the EI (*Figure 1—figure supplement 1B*). We normalized the heights of the somatic expansion peaks to the height of the reference peak, excluding any that were

**Table 1.** Descriptive data on fetal tissues.

| N° | Fetal brain (GW13) | | Trophoblast (GW11) | | Parent blood | |
|---|---|---|---|---|---|---|
| | CAG | Expansion index | CAG | Expansion index | CAG | Expansion index |
| 1 | 40 | 0.0441 | 40 | 0.0407 | 40 | 0.1255 |
| 2 | 41 | 0.0494 | 41 | 0.0495 | 40 | 0.1073 |
| 3 | 42 | 0.0434 | 42 | 0.0473 | 42 | 0.2383 |
| 4 | 43 | 0.0489 | 43 | 0.0482 | 45 | 0.3434 |
| 5 | 45 | 0.0533 | 45 | 0.0516 | 45 | 0.3692 |
| 6 | 46 | 0.0523 | 46 | 0.0571 | 45 | 0.3692 |
| 7 | 46 | 0.0599 | 46 | 0.0589 | 45 | 0.2390 |
| Summary | 43.3 ± 2.4 | 0.0502 ± 0.006 | 43.3 ± 2.4 | 0.0505 ± 0.006 | 42.8 ± 2.5 | 0.2560 ± 0.1103 |

GW13: gestational week 13. GW11: gestational week 11.

Pearson's correlation between EI and (CAG)n for fetal brain: R = 0.8244, $R^2$ = 0.6796, p=0.023.

less than 3% of the main peak height. We multiplied each peak's height by its position to account for the increased repeat length, then summed the peak heights for each sample. An EI of 0 indicates no expansion beyond the inherited allele, and an index >0 indicates mosaicism of the CAG repeat expansion in the tissue.

The CAG expansion is usually followed by a CAACAG cassette that can be duplicated or, in some cases, deleted (*Ciosi et al., 2019*). There are 21 CAG repeats in the reference sequence NG_009378.1, the cassette CAACAGCCGCCA followed by seven CCG and two CCT. When the cassette is changed to CAGCAGCCGCCA by loss of the CAA interruption (*Wright et al., 2019*), the tract becomes less stable and more prone to expansion (*Khristich and Mirkin, 2020*; *Rolfsmeier et al., 2000*; *Xu et al., 2020*). We did not detect this variant in our samples. We excluded one patient from the original cohort who had an additional CAA interruption in the CAG expansion.

## The somatic expansion index increases over the life span in both blood and brain samples

Because of a long period of clinical prospective follow-up of HD patients at the Pitié-Salpêtrière Hospital, we were able to analyze blood samples that were collected during clinical visits at different ages for 50 HD patients (31 women, 19 men; mean reference (CAG)n 44.6 ± 3.5 [range 39–54]) (*Table 2*). With up to three samples (n = 50 for t1 and t2, n = 12 for t3), taken on average 12 and 7 years apart, respectively, we could analyze the progression of somatic instability over quite a long period of time. The EI increased over time (*Figure 1A*), with the aggregate EI increasing from t1 (0.620 ± 0.655) to t2 (0.881 ± 0.929) to t3 (0.967 ± 0.841) (*Table 2*). Regression and Pearson's correlation showed a significant linear relationship between EI and reference (CAG)n in the blood at t1 (r = 0.816, slope = 0.155, p=5.0e-13), t2 (r = 0.880, slope = 0.237, p<2.2e-16), and t3 (r = 0.901, slope = 0.203, p=6.3e-5) (*Figure 1B*, left). It is interesting to note that in our cohort, the lowest index value associated with a symptomatic subject was 0.137; this patient had a reference repeat of 39 CAGs and showed overt motor signs at the age of 49 (*Table 2*).

We were able to evaluate cortices from a separate group of 15 deceased patients (*Figure 1B*, right). As expected from previous studies (*Shelbourne et al., 2007*; *Telenius et al., 1994*), these tissues had the highest EI in our cohort (3.361 ± 2.390, range: 1.288 to 9.094) (*Table 3*), which correlated with the CAG repeat length (r = 0.615, slope = 0.492, p=0.015). We also had a post-mortem brain from a juvenile-onset case with a reference CAG repeat size of 128. The extreme mosaicism in this tissue, however, made it difficult to determine a main CAG peak or calculate an EI using the PCR profile, so we did not include it in our analyses (*Figure 1—figure supplement 1C*).

Because severe neuronal loss could skew the detection of expansions (*Mouro Pinto et al., 2020*), we were particularly interested in examining brain tissue from early development. We analyzed fetal cortical samples from seven HD gene carriers at 13 weeks' gestation (CAG: 40–46, *Table 1*; *Figure 1B,C*; *Barnat et al., 2020*). Although the adult HD cortex has been consistently found to

**Table 2.** Data on the 50 patients in the longitudinal study, arranged according to reference CAG repeat length.

| | | | First sample | | | | Second sample | | | | Third sample | | | | | | | |
|----|-----|-------------|-----|--------|-------|--------|-----|--------|-------|--------|-----|--------|-------|--------|---------|--------|--------------|----|
| N° | CAG | Motor onset | Age | Status | UHDRS | EI | Age | Status | UHDRS | EI | Age | Status | UHDRS | EI | ER | EI-AO | AO Group | AD |
| 1 | 39 | 49 | 39 | P | 0 | 0.1187 | 50 | M | 7 | 0.1369 | | | | | 0.00166 | 0.1355 | Earlier | |
| 2 | 40 | 58 | 40 | P | 0 | 0.1591 | 58 | M | 17 | 0.1910 | | | | | 0.00177 | 0.1908 | Later | |
| 3 | 40 | 80 | 76 | M | 0 | 0.2048 | 87 | M | 36 | 0.2499 | | | | | 0.00409 | 0.2210 | Later | 91 |
| 4 | 41 | 44 | 46 | M | 0 | 0.2120 | 65 | M | 34 | 0.3970 | 69 | M | 75 | 0.4265 | 0.00946 | 0.1941 | Earlier | 77 |
| 5 | 41 | 37 | 28 | M | 0 | 0.2929 | 42 | M | 27 | 0.3806 | 45 | M | 42 | 0.4274 | 0.0074 | 0.2905 | Earlier | |
| 6 | 42 | 39 | 32 | P | 0 | 0.1688 | 46 | M | 28 | 0.2173 | 53 | M | 18 | 0.3446 | 0.00767 | 0.2079 | Earlier | |
| 7 | 42 | 42 | 32 | P | 0 | 0.1968 | 39 | P | 0 | 0.2917 | 51 | M | 25 | 0.3596 | 0.00826 | 0.2939 | Earlier | |
| 8 | 42 | 43 | 34 | P | 0 | 0.1991 | 47 | M | 31 | 0.3387 | | | | | 0.01074 | 0.2959 | As expected | |
| 9 | 42 | 45 | 39 | P | 0 | 0.2145 | 52 | M | 22 | 0.2380 | | | | | 0.00181 | 0.2255 | As expected | |
| 10 | 42 | 48 | 50 | M | 21 | 0.2982 | 59 | M | 76 | 0.3596 | 61 | M | 76 | 0.3264 | 0.00386 | 0.2943 | As expected | 66 |
| 11 | 42 | 50 | 51 | M | 18 | 0.1943 | 59 | M | 39 | 0.2606 | | | | | 0.00829 | 0.1859 | As expected | |
| 12 | 42 | 50 | 50 | P | 4 | 0.4070 | 61 | M | 40 | 0.4239 | | | | | 0.00154 | 0.4072 | As expected | |
| 13 | 42 | 60 | 62 | M | 45 | 0.3774 | 73 | M | 78 | 0,4249 | | | | | 0.00432 | 0.3685 | Later | 74 |
| 14 | 42 | 57 | 56 | M | 0 | 0.3981 | 70 | M | 70 | 0.4905 | | | | | 0.0066 | 0.3980 | Later | |
| 15 | 42 | 56 | 60 | M | 20 | 0.4311 | 68 | M | 45 | 0.4581 | | | | | 0.00338 | 0.4176 | Later | 75 |
| 16 | 42 | 61 | 63 | M | 36 | 0.4569 | 65 | M | 53 | 0.4563 | | | | | −0.0003 | 0.4576 | Later | |
| 17 | 43 | 37 | 29 | P | 0 | 0.1806 | 53 | M | 31 | 0.3641 | | | | | 0.00765 | 0.2418 | Earlier | |
| 18 | 43 | 45 | 43 | P | 0 | 0.3159 | 59 | M | 40 | 0.4275 | | | | | 0.00698 | 0.3299 | As expected | |
| 19 | 43 | 45 | 37 | P | 0 | 0.2879 | 52 | M | 38 | 0.4628 | 54 | M | 43 | 0.5088 | 0.01249 | 0.3865 | As expected | |
| 20 | 43 | 47 | 46 | P | 0 | 0.3255 | 56 | M | 32 | 0.4228 | | | | | 0.00973 | 0.3351 | As expected | 64 |
| 21 | 43 | 47 | 50 | M | 18 | 0.4430 | 58 | M | 57 | 0.5164 | | | | | 0.00917 | 0.4153 | As expected | |
| 22 | 44 | 48 | 21 | P | 0 | 0.1731 | 29 | P | 0 | 0.2063 | 47 | M | 18 | 0.2866 | 0.00438 | 0.2862 | Later | |
| 23 | 44 | 56 | 39 | P | 0 | 0.2002 | 62 | M | 38 | 0.4136 | | | | | 0.00928 | 0.3581 | Later | |
| 24 | 44 | 44 | 32 | P | 0 | 0.2176 | 35 | P | 0 | 0.2214 | | | | | 0.00127 | 0.2327 | As expected | 56 |
| 25 | 44 | 44 | 25 | P | 0 | 0.2162 | 42 | M | 12 | 0.5379 | | | | | 0.01892 | 0.5377 | As expected | |
| 26 | 44 | 37 | 31 | P | 0 | 0.3739 | 44 | M | 32 | 0.5477 | | | | | 0.01337 | 0.4541 | Earlier | 49 |
| 27 | 44 | 44 | 37 | P | 0 | 0.3635 | 50 | P | 0 | 0.4687 | | | | | 0.00809 | 0.4201 | As expected | 57 |
| 28 | 44 | 37 | 28 | P | 0 | 0.3751 | 43 | M | 35 | 0.5175 | | | | | 0.00949 | 0.4604 | Earlier | |
| 29 | 44 | 50 | 42 | P | 0 | 0.4229 | 54 | M | 38 | 0.5162 | | | | | 0.00778 | 0.4851 | Later | |
| 30 | 44 | 26 | 26 | M | 11 | 0.4279 | 36 | M | 22 | 0.5113 | | | | | 0.00834 | 0.4279 | Earlier | |
| 31 | 44 | 46 | 49 | M | NA | 0.4550 | 60 | M | 86 | 0.5854 | | | | | 0.01185 | 0.4193 | As expected | 61 |
| 32 | 44 | 30 | 36 | P | 0 | 0.4272 | 50 | M | 49 | 0.5667 | | | | | 0.00997 | 0.3674 | Earlier | 54 |

*Table 2 continued on next page*

*Table 2 continued*

| N° | CAG | Motor onset | First sample | | | | Second sample | | | | Third sample | | | | ER | EI-AO | AO Group | AD |
|---|---|---|---|---|---|---|---|---|---|---|---|---|---|---|---|---|---|---|
| | | | Age | Status | UHDRS | EI | Age | Status | UHDRS | EI | Age | Status | UHDRS | EI | | | | |
| 33 | 45 | 37 | 20 | P | 0 | 0.3551 | 38 | M | 7 | 0.5407 | | | | | 0.01031 | 0.5303 | As expected | |
| 34 | 45 | 40 | 35 | P | 0 | 0.4522 | 49 | M | 52 | 0.6794 | 51 | M | 55 | 0.7050 | 0.01596 | 0.5325 | As expected | |
| 35 | 45 | 40 | 46 | M | NA | 0.7408 | 58 | M | NA | 1.0457 | | | | | 0.02541 | 0.5884 | As expected | |
| 36 | 46 | 40 | 27 | P | 0 | 0.2948 | 48 | M | 43 | 0.6126 | | | | | 0.01514 | 0.4917 | As expected | |
| 37 | 46 | 36 | 26 | P | 0 | 0.5122 | 39 | M | 58 | 0.7044 | | | | | 0.01478 | 0.6599 | As expected | |
| 38 | 46 | 35 | 35 | P | 0 | 0.6031 | 45 | M | 91 | 0.7675 | | | | | 0.01645 | 0.6031 | As expected | 49 |
| 39 | 46 | 45 | 47 | M | 18 | 0.7701 | 55 | M | 59 | 0.9431 | | | | | 0.02163 | 0.7270 | Later | |
| 40 | 47 | 36 | 46 | M | 45 | 1.5687 | 57 | M | 92 | 2.6750 | | | | | 0.10057 | 0.5629 | As expected | 64 |
| 41 | 48 | 30 | 21 | P | 0 | 0.7800 | 38 | M | 57 | 1.6341 | | | | | 0.05024 | 1.2321 | Earlier | 39 |
| 42 | 48 | 32 | 33 | M | 8 | 0.9630 | 43 | M | 84 | 1.2162 | 53 | M | 92 | 1.8055 | 0.04213 | 0.8650 | As expected | |
| 43 | 49 | 39 | 29 | P | 0 | 1.2709 | 42 | M | 84 | 1.7097 | 42 | | | 1.7208 | 0.03419 | 1.6129 | Later | |
| 44 | 49 | 27 | 20 | P | 0 | 0.8439 | 27 | M | 63 | 1.3535 | | | | | 0.0728 | 1.3534 | Earlier | |
| 45 | 49 | 33 | 41 | M | 43 | 1.6512 | 43 | M | 53 | 1.9788 | 51 | M | 70 | 2.6855 | 0.09912 | 0.9159 | As expected | |
| 46 | 50 | 25 | 31 | M | NA | 2.7278 | 41 | M | NA | 3.6178 | | | | | 0.089 | 2.1936 | Earlier | |
| 47 | 52 | 25 | 20 | P | 0 | 0.8479 | 33 | M | 32 | 1.7060 | 39 | M | 69 | 2.0014 | 0.06156 | 1.1696 | Earlier | |
| 48 | 52 | 27 | 36 | M | NA | 3.1542 | 36 | M | 49 | 3.3273 | | | | | NA | NA | As expected | |
| 49 | 53 | 26 | 26 | M | 15 | 1.9819 | 34 | M | 22 | 3.0756 | | | | | 0.13672 | 1.982 | As expected | |
| 50 | 54 | 36 | 20 | P | 0 | 1.9449 | 36 | M | 13 | 3.8745 | | | | | 0.1206 | 3.8745 | Later | |
| | 44.6 ± 3.5 | 42.2± 10.8 | 37.8± 12.6 | 19 M/ 31P | 6.6± 12.9 | 0.620± 0.655 | 49.7 ± 12.3 | 46 M/ 4P | 41.1± 25.1 | 0.881± 0.929 | 51.3± 8.1 | 12M/0P | 53± 25.6 | 0.967 ± 0.841 | 0.0236± 0.0332 | 0.625± 0.649 | | 62.9± 13.6 |

UHDRS: United Huntington's Disease Rating Scale/124; P: premanifest; M: manifest; ER : expansion rate; EI: expansion index; AD: age at death; AO: age at onset.

bear the greatest somatic expansions, the fetal cortex showed almost no mosaicism: the somatic EIs were very small, ranging from 0.043 to 0.060 (0.050 ± 0.006), though they still correlated with CAG repeat length (p=0.023) (*Table 1*). These indices were extremely close to those from trophoblast tissues that were analyzed for prenatal diagnosis between 11 and 12 weeks' gestation (*Figure 1C*, left). Yet blood samples taken from their premanifest carrier parents at the same time (n = 6, CAG: 42.8 ± 2.5, 40–45; *Table 1*; these adults were not part of the longitudinal cohort) showed somatic expansions, with a mean EI of 0.256 ± 0.11 (range: 0.107 to 0.369; *Figure 1C*, left).

To better visualize these differences between parental blood and fetal tissue, we graphed somatic mosaicism in fetal cortices, trophoblasts, and premanifest parents for four different reference CAG lengths and estimated the percent of mutant alleles harboring each somatic expansion length (*Figure 1C*, right). There is clearly more variability in the parental blood (dark orange bars) than in the fetal brain tissue (green bars). Similarly, comparison of somatic mosaicism in three of the fetal brains, the blood samples (across three timepoints) from three patients in our longitudinal

**Table 3.** Descriptive data on post-mortem brain donors.

| N° | Sex | CAG | Expansion index | Age at death |
|---|---|---|---|---|
| 1 | F | 40 | 1.4506 | 74 |
| 2 | F | 41 | 1.3197 | 84 |
| 3 | F | 42 | 3.2444 | 75 |
| 4 | M | 43 | 1.6834 | 55 |
| 5 | M | 43 | 2.6007 | 68 |
| 6 | F | 44 | 1.2883 | NA |
| 7 | M | 44 | 3.2551 | 41 |
| 8 | F | 44 | 3.0440 | 59 |
| 9 | F | 44 | 3.7083 | 55 |
| 10 | F | 45 | 2.9722 | 59 |
| 11 | F | 46 | 1.5191 | 43 |
| 12 | F | 47 | 9.0942 | 43 |
| 13 | M | 48 | 2.8506 | 41 |
| 14 | F | 50 | 3.7961 | 58 |
| 15 | F | 50 | 8.5912 | 45 |
| Summary | 11F/5M | 44.7 ± 3.0 | 3.3612 ± 2.3900 | 57.4 ± 13.9 |

cohort, and three adult post-mortem cortices (*Figure 2*) clearly shows that mosaicism increased over time in blood cells but was even more marked in the adult brain, with more additional CAGs for a given reference CAG repeat size.

## Determination of somatic expansion rate

We next asked whether the propensity to expand grows over time, and whether an 'expansion rate' (ER) that estimates the average annual expansion growth for each patient would correlate with the available clinical outcomes. To this end, we first ruled out the possibility of a sex effect by verifying that there was no sex difference in the AO (female: n = 31, 41.9 ± 8.5 [range 25–61]; male: n = 19, 42.4 ± 14.3 [range 25–80]; Wilcoxon rank-sum test, p=0.899) or in the age at death (AD) (female: n = 7, 59.4 ± 9.9 [range 49–77]; male: n = 7, 65.7 ± 16.7, 39–91; Wilcoxon rank-sum test, p=0.442).

We then calculated an ER for each of the 50 subjects using the slope of the regression line for the EI on ages at visits (0.024 ± 0.033 units per year [range −0.0003 to 0.1367], *Table 2*). Because calculating a rate entails having a baseline, we chose to extrapolate a plausible, if theoretical, EI at AO (EI-AO). To do so we used the slope and the intercept (estimated EI at birth) for each patient to estimate EI-AO (see 'Materials and methods'). A Pearson's correlation coefficient of r = 0.861 (p=2.1e-15) showed a strong association between the reference CAG repeat size and the somatic ER (*Figure 3—figure supplement 1*). Also, a Pearson's correlation coefficient of r = 0.847 (p=1.6e-14) showed a strong association between the reference CAG repeat size and EI-AO (*Figure 3—figure supplement 1*).

## EI and ER correlations with age at onset, age at death, and disease manifest status

To determine whether EI or ER could explain the variation in AO not explained by the reference repeat, we first needed to calculate how (CAG)n correlates with AO in our sample. In our longitudinal cohort of 50 subjects, (CAG)n accounted for 47.6% of variance in AO, which is at the low end of the published ranges (~42–71%) (*Squitieri et al., 2006*; *Figure 3A*, left). This is likely due to our small sample relative to many such studies, which can include hundreds to thousands of patients. CAG repeat length accounted for 68% of variance in age at death (AD) (*Figure 3A*, right). Nevertheless, we proceeded to analyze the relationships between EI-AO, ER, AO, and AD. EI-AO had an inverse correlation with AO (r = −0.437, p=1.7e-03) and AD (r = −0.666, p=9.3e-03) (*Figure 3—figure supplement 1*) and accounted for 20.7% of the variance in AO and 49.7% of variance in AD

**Table 4.** Descriptive data on the Linear mixed model analyzing the longitudinal data.
Linear mixed model fit by REML. t-tests use Satterthwaite's method ['lmerModLmerTest']. Formula:
log(ei)~age + cag+age:cag +sex + (1 | id).

| Outcome | Fixed effects | | | | | | Signif | Random effects | | | |
|---|---|---|---|---|---|---|---|---|---|---|---|
| | | Estimate | Std. Error | df | t value | Pr(>\|t\|) | | Groups | Name | Variance | Std. Dev. |
| log(ei) | (Intercept) | −0.603 | 0.048 | 48.31 | −12.5 | 1.03e-16 | *** | id | (Intercept) | 0.067 | 0.258 |
| | age | 0.028 | 0.001 | 70.39 | 22.8 | 3.40–34 | *** | Residual | | 0.009 | 0.095 |
| | cag | 0.276 | 0.012 | 54.21 | 24.0 | 1.63e-30 | *** | Number of obs: 112, groups: id, 50 | | | |
| | sexMale | 0.028 | 0.078 | 47.03 | 0.36 | 0.719 | ns | | | | |
| | age:cag | 0.002 | 3.8e-4 | 77.22 | 5.69 | 2.21e-07 | *** | | | | |

Signif: ***p<0.001, $^{ns}$p >0.05.

from the longitudinal group (n = 14 patients who died during the study) (*Figure 3B*). ER had an inverse correlation with AO (r = −0.541, p=5.9e-05) and AD (r = −0.261, p=3.7e-01) (*Figure 3—figure supplement 1*); it accounted for 33% of the variance in AO and did not account for the variance in AD from the longitudinal group (*Figure 3C*). Notably, ER explained a larger proportion of AO variance than EI-AO. EI-AO accounted for more of the variance in AD than did ER, but it is difficult to draw conclusions based on the small sample of patients for the AD data.

We then took an alternative approach to understanding variation in AO. We classified individuals into three groups indicating expected AO, earlier- or later-than-expected AO, as defined by the model errors in the linear regression of AO and reference CAG repeat size (*Figure 3A*, left; see 'Materials and methods'). Neither EI-AO nor ER accounted for the differences in AO among these groups, despite a trend for lower ER in the later-than-expected group (Kruskal-Wallis test, rate: p=0.181, EI-AO: p=0.810, *Figure 4A*). Given the difficulties inherent in pinpointing AO, we asked whether we could see an influence of residual ER on the more general classification of premanifest vs manifest. Here we found significant differences between groups in both residual EI and residual ER (Wilcoxon test, p=3.5e-05 and p=0.023 respectively, *Figure 4B*).

We next asked whether we could find correlations based on the post-mortem cortices (*Table 3*). A previous study on post-mortem HD brains showed that, after accounting for reference CAG repeat size, greater somatic expansions in the cortex correlated significantly with earlier AO (*Swami et al., 2009*). Since we did not have information on AO for the 14 subjects in the post-mortem group, we asked whether EI (from the postmortem samples) or ER (from the blood sample group) correlated with residual AD. We calculated the residual AD after accounting for the effect of the reference CAG repeat length, compared to the ER derived from the blood measures (p=0.028, $R^2_{adj}$ = 28.6%; *Figure 4—figure supplement 1A*) or to the EI derived from the postmortem cohort (p=0.578, $R^2_{adj}$ < 0, *Figure 4—figure supplement 1B*). With the caveat that we do not know the cause of death in all cases (which could be due to causes other than HD), EI from brain samples did not correlate with the residual AD, but ER from blood samples correlated weakly with residual AD. A larger sample would likely reveal stronger correlations.

## Within-subject variation in somatic mosaicism depends on (CAG)n, age, and the interaction of these two variables

We next sought to understand the relative contributions of the reference repeat size and age on the tendency toward somatic expansions. To account for the repeated measurements for each patient, a linear mixed-effects model (LMM) was fitted to the EI data on a log scale. Based on the fixed effects of the derived model, we found significant effects of age (coefficient = 0.028, SE = 0.001, p=3.4e-34) and number of CAG repeats (coefficient = 0.276, SE = 0.012, p=1.6e-30). In addition, the significant interaction between age and CAG repeat length suggests that, as CAG repeat length increases, the expansions become greater each year (coefficient = 0.002, SE = 3.8e-4, p=2.2e-7) (*Table 4*). As

both age and CAG were mean-centered in the model, the exponential intercept would also indicate a predicted EI of 0.547 (intercept = −0.603, SE = 0.048, p=1.0e-16) for a hypothetical patient carrying the mean characteristics of the cohort (i.e., average age in the cohorts of 44.6 years and mean CAG repeat size of 44.7). Sex was again used as a cofactor and did not show any significant effect on EI (coefficient = 0.028, SE = 0.078, p=0.719).

Finally, the contribution of each fixed effects term explaining the EI, given by t values (estimate divided by SE) in descending order of importance, was as follows: t(CAG)=24.0, t(age)=22.8 and t (age ×CAG)=5.7. Based on the fixed effects estimation extracted from the LMM, we plotted trajectories for the EI as of function of age (one trajectory for each CAG repeat length). The predicted values of each EI are shown on the original scale after back-transformation from the logarithmic scale over the same age intervals from the patient cohort for each (CAG)n (*Figure 5*). This model provides a glimpse of how instability evolves with (CAG)n, age, and the interaction between these two factors.

## Discussion

Our longitudinal study provides data that support the Kaplan model in several ways (*Kaplan et al., 2007*). First, the model predicts that at the beginning of life, all disease-relevant cells begin with the inherited repeat and negligible somatic instability. This turns out to be the case: we found almost no somatic mosaicism at the fetal stage. One might have expected that the high number of mitoses at this stage of brain development would make neural precursors sensitive to double-strand breaks and replication errors (*Leija-Salazar et al., 2018*; *Schwer et al., 2016*), but somatic expansions occur through different mechanisms than germline expansions (*Khristich and Mirkin, 2020*; *Tabrizi et al., 2020*). Although we did not have samples from embryos at later stages, there is such data for other diseases caused by repeat expansions. For instance, in Friedreich ataxia, which is caused by an expanded GAA repeat in the first intron on both alleles of the *FXN* gene, levels of instability found in tissues from an 18-week-old fetus were very low compared to adult-derived tissues (*De Biase et al., 2007*). In myotonic dystrophy, caused by a non-coding CTG repeat expansion in the *DMPK* gene, repeat instability was not observed at 13 weeks in fetal tissues, but a difference between tissues became detectable after 16 weeks (*Martorell et al., 1997*). All these studies suggest that, early in life, somatic instability is minimal.

Second, Kaplan et al. posited that somatic expansions should progress with age, even prior to disease onset. This also turns out to be correct: the presymptomatic carrier parents of fetal mutation carriers already showed somatic instability in the blood at the time of the pregnancy. It is remarkable, in fact, that increases in ER were evident despite our limited sample size and despite the fact that we had to derive this calculation from blood cells, which are not involved in HD pathogenesis and completely change over every six months or so. Unfortunately, for this very reason, the EI from the blood is not sufficient to predict AO, which is influenced by not only repeat length and somatic instability but other factors such as variants in DNA repair factors (see below).

Third, the model predicts the rate of allele expansion should increase with time and be a function of the repeat length at that time. This is indeed what we found: not just greater somatic expansions with age and reference repeat length (as represented by EI), but a greater propensity to expand with age (as represented by ER).

One prediction of the Kaplan model we could not test is that there should be different thresholds of somatic expansions that must be reached for different brain regions to become pathological. It is hard to imagine how this particular prediction of the model could be tested, other than by performing extensive neuropathological studies on a great many mice at many different disease stages. In terms of correlation between somatic instability and disease progression, we did find group-level differences in EI and ER between the premanifest and manifest state. We could not establish a correlation at the individual subject level, however, likely because of the limited sample size as well as the difficulty of pinpointing phenoconversion in a disease that continues to unfold over many years. Stronger evidence on this point came from a large study of nearly 750 HD mutation carriers, which showed that larger somatic expansions are associated with worse clinical outcomes (earlier AO, higher motor and progression scores) in HD (*Ciosi et al., 2019*).

The most interesting questions that remain to be answered have to do with what drives somatic instability. The brain regions that have the greatest repeat expansions in HD, the striatum and

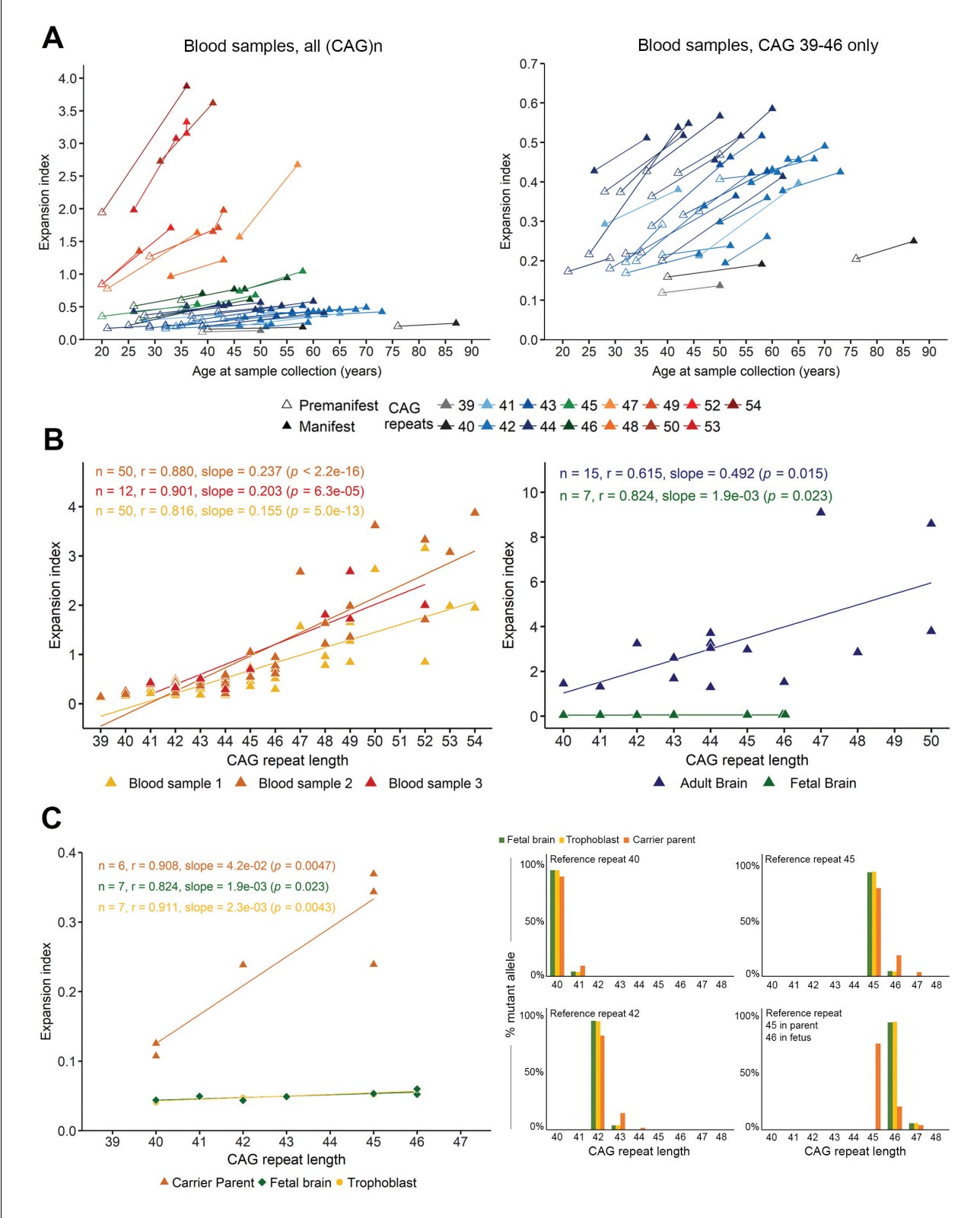

**Figure 1.** Somatic instability is negligible during gestation but increases with age. (**A**) Left panel: Changes in expansion index (EI) for each Huntington disease (HD) individual, across two or sometimes three visits over multiple years (see **Table 5**). Different colors indicate different reference (CAG)n at diagnosis (main peak on PCR profile as shown in **Figure 1—figure supplement 1**). For each value, the disease status is indicated with an empty triangle (premanifest) or filled triangle (manifest; score >5 on the Unified HD Rating Scale total motor score [UHDRS-TMS]). Note that the measurement

*Figure 1 continued on next page*

*Figure 1 continued*

of the inherited allele remained stable across visits except for patient 44 (**Table 2**), who went from 54 to 55 (CAG)n at the second sampling, which was taken into account for the EI. Right panel: A closer look at the values clustered at the bottom of the axis in **A** ((CAG)n 39–46). The EI increases with progression to manifest state even for individuals with relatively small reference repeats. (**B**) Scatter plot and regression lines show the linear relation between EI and (CAG)n as observed in adult blood from the longitudinal cohort (left panel, n = 50 patients with at least two samples each; of these, 12 had a third sample; yellow: first sample, orange: second sample, and red triangles: third sample) and cortical tissue (right panel, n = 7 fetal brains, green triangles, and n = 15 adult brains, blue triangles). Pearson's correlation coefficients and estimated regression slopes with p-values, indicated in the upper portion of each graph, reveal a positive linear relation between EI and reference CAG repeat length. (**C**) EI values from seven fetal samples according to the reference CAG repeat, ranging from 40 to 46 (at 13 weeks gestation). The instability indices of the cortical samples (green) overlap with those of the trophoblast samples (yellow); indices from carrier parents' blood are in orange (ages 25 to 34 years). Two of the fetuses had the same (CAG)n of 46 and thus overlap on the graph. Left panel: Comparison of brain tissue instability from HD carrier fetuses at 13 weeks (green), to the corresponding trophoblasts sampled for prenatal testing (yellow) and the premanifest carrier parents' blood (orange). Right panel: The percentage of mutant alleles bearing the different somatic expansions ascertained from the peak heights. Four graphs were plotted for the reference CAGs (40, 42, 45, and 46) determined on the fetal tissues. The parental blood samples show significant somatic expansions, whereas the trophoblast and the developing cortex show very little.

The online version of this article includes the following figure supplement(s) for figure 1:

**Figure supplement 1.** PCR profile and analysis of the CAG repeat length and instability.

cortex, are hypermetabolic from early in the disease course (*Tereshchenko et al., 2020*), and neurons show greater somatic instability than glial cells in models and post mortem brains (*Gonitel et al., 2008*; *Shelbourne et al., 2007*). Metabolic stress may also lead to mitochondrial dysfunction and energy deficit in HD (*Mochel et al., 2012*; *Roze et al., 2008*; *Tabrizi et al., 1999*). An excess of excitatory glutamatergic inputs and NMDA receptor activation creates energy demands that are not sustainable in a context of diminished energy capacity, and may lead to cell death (*Milnerwood et al., 2010*; *Mochel and Haller, 2011*).

In fact, an excitotoxicity model of neurodegeneration was proposed for HD many years before the discovery of the genetic basis of the disease (*Coyle and Schwarcz, 1976*; *Mcgeer and Mcgeer, 1976*). The medium spiny neurons of the caudate and putamen, which are the most vulnerable in HD, receive their main input from cortical glutamatergic neurons; they are thus particularly

**Table 5.** Descriptive data on longitudinal cohort.

| | t1 | t2 | t3 |
|---|---|---|---|
| n | 50 | 50 | 12 |
| Sex, F/M (%F) | 31/19 (62%) | 31/19 (62%) | 7/5 (58.3%) |
| CAG | 44.6 ± 3.5 (n = 50, r = 39–54) | NA | NA |
| Expansion index | 0.620 ± 0.655 (n = 50, r = 0.119–3.154) | 0.881 ± 0.929 (n = 50, r = 0.137–3.874) | 0.967 ± 0.841 (n = 12, r = 0.287–2.686) |
| Age at sampling | 37.8 ± 12.6 (n = 50, r = 20–76) | 49.7 ± 12.3 (n = 50, r = 27–87) | 51.3 ± 8.1 (n = 12, r = 39–69) |
| Status M/P (%M) | 19/31 (38%) | 46/4 (92%) | 12/0 (100%) |
| Chorea onset yes/no (%yes) | 20/30 (40%) | 46/4 (92%) | 11/1 (91.7%) |
| UHDRS | 6.6 ± 12.9 (n = 46, r = 0–45) | 41.1 ± 25.1 (n = 48, r = 0–92) | 53 ± 25.6 (n = 11, r = 18–92) |
| | **t2 - t1** | | **t3 - t2** |
| Delta age | 12 ± 4.9 (n = 50, r = 0–24) | | 6.2 ± 5.2 (n = 12, r = 0–18) |
| Delta expansion | 0.16 ± 0.22 (n = 50, r = –0.01–1.37) | | 0.11 ± 0.12 (n = 12, r = –0.03–0.37) |
| Rate (delta expansion/delta age) | 0.0136 ± 0.016 (n = 49, r = –0.0036–0.0859) | | 0.0182 ± 0.0182 (n = 11, r = –0.0153–0.0461) |
| Expansion rate (ER) | 0.0236 ± 0.0332 (n = 49, r = –0.0003–0.1367) | | |
| Expansion index at onset (EI-AO) | 0.625 ± 0.649 (n = 49, r = 0.136–3.874) | | |

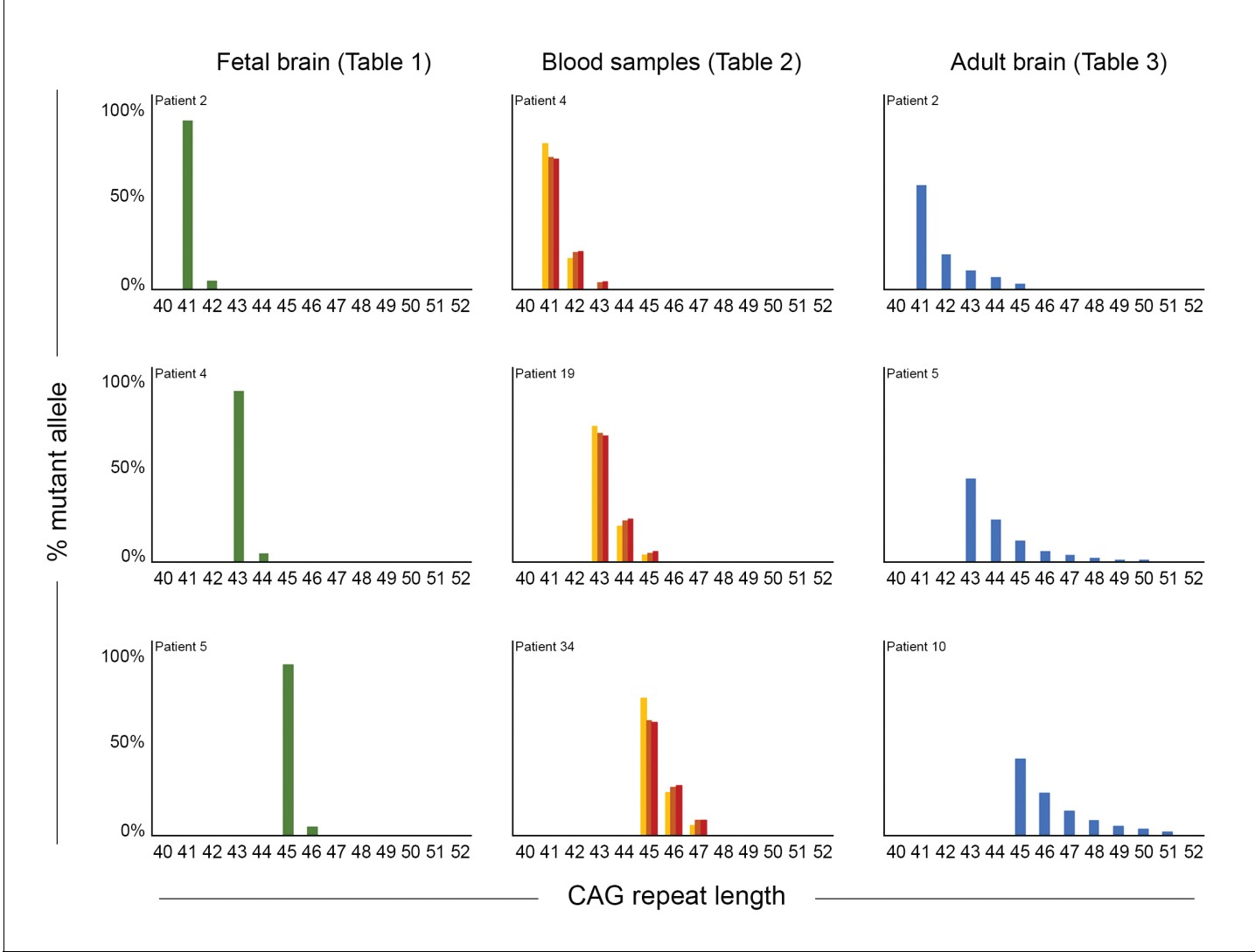

**Figure 2.** Somatic mosaicism increases in blood and post-mortem cortex over time. Comparison of mosaicism in cortical tissue from Huntington disease (HD) carrier fetuses at 13 weeks (green), blood samples over time (gold, orange, and red for t1, t2, and t3, respectively) and adult post-mortem cortices (blue). We ascertained the '% mutant alleles' (as in *Figure 1C*) from the peak heights from PCR profiles obtained on GeneMapper. Three reference CAG lengths (41, 43, or 45 CAG) were chosen from our cohort to illustrate the evolution of instability, and each graph represents one individual patient (from top to bottom): for the fetal samples, patients 2, 4, and 5 (*Table 1*); for the blood samples, the repeated measures from patients 4, 19, and 34 (*Table 2*); for the post-mortem samples, patients 2, 5, 10 (*Table 3*).

susceptible to excitation and, in fact, HD can be mimicked by administering glutamate analogues to the striatum (*Coyle and Schwarcz, 1976*; *Estrada Sánchez et al., 2008*; *Mcgeer and Mcgeer, 1976*). In this context it is worth noting that variants in the GluR6 kainate receptor locus were found to account for 13% of variation in AO that was not provided by CAG repeat number (*Rubinsztein et al., 1997*). Along similar lines, a recent study showed that absence of the aryl hydrocarbon receptor (AhR), which protects mice from excitotoxicity, greatly reduced behavioral deficits in the R6/1 transgenic model of HD (*Angeles-López et al., 2021*).

Hypermetabolism would also contribute to oxidative stress, which can cause DNA damage (*Iyer and Pluciennik, 2021*; *Leija-Salazar et al., 2018*). Large-scale studies have linked somatic CAG expansions in patients' blood to the presence of variants in DNA repair genes, not just in HD (*Ciosi et al., 2019*; *Lee et al., 2019*) but in other polyglutamine diseases as well (*Bettencourt et al., 2016*). In HD, somatic instability is influenced by polymorphisms in *MSH3, MLH1, MlH3,* and *FAN1*, which are all involved in DNA repair (*Ciosi et al., 2019*). Counterintuitively, loss of function of some

**B**

**C**

**Figure 3.** Somatic expansion correlates with reference CAG repeat length, with age at onset (AO) and age at death (AD). Scatter plots showing (**A**) reference CAG repeat length, (**B**) expansion index (EI) estimated at age at onset (EI-AO) and (**C**) expansion rate (ER) with respect to AO (left panels) and AD (right panels). p-Values of the slopes and adjusted R-squared for the linear regressions of log(AO) and log(AD) are in the upper right corner of the plots. Red curves denote locally estimated scatterplot smoothing (LOESS) of the data with 95% confidence intervals shaded in gray. (**A**) Left panel: The

*Figure 3 continued on next page*

*Figure 3 continued*

reference CAG repeat length explains roughly half the variability in AO ($R^2_{adj}$ = 47.6%, p=1.8e-8). HD individuals were classified as having onset earlier than expected (green), as expected (grey), or later than expected (red) according to their distance from the linear predictions given by the CAG repeat (see 'Materials and methods'). Right panel: The reference CAG repeat length explains 68% of the variability in AD (p=1.7e-04). (**B**) EI-AO explains 20.7% of the variability in AO (p=6.0e-04, left panel) and 49.7% of the variability in AD (p=2.9e-03, right panel). (**C**) ER explains 33% of the variability in AO (p=9.3e-06, left panel) and does not explain the variability in AD ($R^2_{adj}$ < 0, p=0.348, right panel).

The online version of this article includes the following figure supplement(s) for figure 3:

**Figure supplement 1.** Pearson's correlation analysis.

DNA repair factors can be protective: knockout of *Msh2* or *Msh3* in a knock-in model of HD prevents CAG expansions in the striatum (*Pinto et al., 2013*). The reason for this may be that transcriptionally active genes elicit mismatch repair activity to guard genomic integrity, but long repeat tracts are difficult to repair accurately (*Iyer and Pluciennik, 2021*). A different mechanism is at work for *FAN1*, which actually stabilizes the CAG repeat in HD (*Goold et al., 2019*); loss of *FAN1* function increases repeat instability (*Kim et al., 2020*; *Loupe et al., 2020*). Interestingly, there is evidence that double-strand break repair is dysregulated in HD: ATM (ataxia-telengiectasia mutated) is upregulated in brain tissue from HD mice and patients, and its heterozygous loss of function is protective in both mouse and *Drosophila* models of HD (*Lu et al., 2014*). It could be that the decline in DNA repair capacity or efficiency that comes with age (*Gorbunova et al., 2007*) contributes to the increasing somatic instability in blood cells, which, as we noted above, seem too short-lived to accumulate expansions as they do. An extended longitudinal study of the effect of DNA repair gene variants on somatic instability would be of great interest.

Given that somatic instability influences disease progression, targeting the repeat instability is a very appealing disease-modifying strategy (*Khristich and Mirkin, 2020*). One possibility is to introduce DNA-stabilizing interruptions into the repeat tract via gene editing (*Ciosi et al., 2019*). Another is to modulate DNA repair activity in HD to retard somatic expansions (*Dragileva et al., 2009*; *Pinto et al., 2013*), but this might also run the risk of increasing overall genomic instability. A recent approach using a small molecule that specifically binds CAG slip-out structures was able to contract the expansions and reduce protein aggregates in the striatum of R6/2 mice (*Nakamori et al., 2020*). Further efforts to stabilize or contract somatic expansions are warranted, particularly if expansions within brain tissue can be reduced. Last but not least, there is much more work to be done to understand the mechanisms that trigger somatic expansions, whether they relate to excitotoxicity, and how they lead to neurodegeneration.

## Materials and methods

### Key resources table

| Reagent type (species) or resource | Designation | Source or reference | Identifiers | Additional information |
|---|---|---|---|---|
| gene (human) | *Htt* | NCBI | NG_009378.1 | |
| sequence-based reagent | HD-F2 | This paper | PCR primers | GGGAGACCGCCATGGCGACCCTGGA |
| sequence-based reagent | HD-WR2-hex | https://doi.org/10.1006/mcpr.1993.1034 | PCR primers | HEXGGCGGTGGCGGCTGTTGCTGCTGCT |
| sequence-based reagent | HD-WCAAM4-R-fam | This paper | PCR primers | [6FAM]GGCGGTGGCGGCTGTTGCTGTTGAT |
| commercial assay or kit | QIAamp Fast DNA Tissue Kit | Qiagen | 51404 | |
| commercial assay or kit | Maxwell RSC Blood DNA kit | Promega | AS1400 | |
| commercial assay or kit | Taq DNA Polymerase | Qiagen | 201205 | |

*Continued on next page*

*Continued*

| Reagent type (species) or resource | Designation | Source or reference | Identifiers | Additional information |
|---|---|---|---|---|
| chemical compound, drug | Genescan-400HD Rox dye size standard | Applied Biosystems | 15829736 | |
| chemical compound, drug | Hi-Di Formamide | Applied Biosystems | 15803570 | |
| software, algorithm | GeneMapper software v5.0 | Applied Biosystems | RRID:SCR_014290 | |
| software, algorithm | R version 3.6.1 | R Development Core Team | RRID:SCR_001905 | https://www.R-project.org/ |

## Sample collection

### Longitudinal study

We recruited HD patients through the Department of Genetics of the Pitié-Salpêtrière University Hospital (Paris, France). Inclusion criteria were a pathological CAG repeat expansion in the *HTT* gene above 38 repeats. Age at disease onset was defined as the presence of a clinically significant movement disorder consistent with HD. We obtained blood samples with written informed consent according to the French legislation (approval from local ethics committees on 19/12/1990, 10/11/1992, followed by the Ethics committee Ile de France II on 30/9/2004 and 18/2/2010). All tested subjects were offered long-term follow-up and signed an informed consent prior to clinical examination and interview. We determined AO by taking the earliest date between self-reported age and motor signs at examination by a neurologist.

### Post-mortem cortical samples

Brain samples were collected as part of a program of 'Brain Donation for Research' (National Neuro-CEB Brain Bank, GIE Neuro-CEB BB-0033–00011). Brains were dissected in the neuropathological department of the Pitié-Salpêtrière University Hospital (Paris, France) to isolate samples from the frontal cortex.

### Fetal samples

Approximately 20% of HD mutation carriers request prenatal diagnosis. After analysis of the fetal DNA, obtained by chorionic villus sampling, if the fetus carries the mutation the parents can request termination of the pregnancy, which is performed by manual vacuum aspiration under general anesthesia. Typically, the termination occurs at gestational week 13. We used standard obstetric protocols in accordance with the French guidelines for clinical practice. Prenatal visits and psychological support were provided for all couples participating, as standard practice, and no additional visits were planned due to participation in this study. The women signed an informed consent during a prenatal visit agreeing to the collection of fetal tissue following the eventual termination of the pregnancy. The study complied with all relevant ethical regulations, with approval from the French Agency of biomedicine (n°PFS17-001; 24/01/2017). The brain tissue analyzed was from the developing cortex.

## DNA extraction

Post-mortem brains and fetal tissues were rapidly frozen and stored at -80°C until DNA extraction. DNA was extracted from brain tissues using the QIAamp Fast DNA Tissue Kit (Qiagen S.A., Courtaboeuf Cedex, France), according to manufacturer's instructions. For blood samples, DNA was extracted using the Maxwell RSC Blood DNA kit, according to manufacturer's instructions (Promega, France EURL). Finally, we measured DNA yields using a NanoDrop 8000 spectrophotometer (ThermoScientific, Illkirch Cedex, France).

**A**

**B**

**Figure 4.** Residual expansion index (EI) and expansion rate (ER) correlate with disease status. (**A**) Boxplots showing the distribution of the EI estimated at age at onset and ER, between the patients' groups classified in *Figure 3A* as having earlier-than-expected, expected, or later-than-expected onset. p-Values for the Kruskal-Wallis test are at the top of each plot; thick horizontal lines indicate the median, diamonds indicate the mean. Earlier and later onset patients had similar ERs compared to the as-expected group (p=0.810 and 0.181). (**B**) Boxplots showing the distribution of the residual EI at the first visit and residual ER, between the patients' groups classified as premanifest and manifest. p-Values for the Wilcoxon test are at the top of each plot.

The online version of this article includes the following figure supplement(s) for figure 4:

**Figure supplement 1.** Comparison of the relation between age at death, CAG repeat length, and expansion index (EI) as measured in blood samples from deceased longitudinal patients and in post-mortem HD brains.

**Figure 5.** Evolution of somatic expansions in HD patient blood is a function of age, CAG repeat length, and the interaction between age and repeat length. A linear mixed model was fitted to the longitudinal data using all blood samples collected. The fitted lines show that the predicted somatic expansion increases with age for all reference repeat sizes, most notably at greater reference repeat lengths. The sex of the patients is indicated by solid and dashed lines (male and female respectively). Each curve covers the same age interval observed in our cohort for a given repeat length.

## Measuring somatic CAG repeat expansions and calculating the somatic expansion index

We used the GeneMapper software v5.0 (Applied Biosystems) to analyze the somatic CAG repeat expansions. For any individual, the majority of PCR products peak around a main signal representing the reference CAG repeat size. Signals to the left of this peak include PCR 'stutter' inherent in the assay, but PCR products to the right represent somatically expanded CAG repeats only; these latter peaks were included. From the GeneMapper 'sample plot view,' we exported a data table for each sample containing the following information: sample name, called CAG allele, peak size in base pair (bp), peak height, area under the peak, and data point/scan number of the highest point of the peak. Based on the main expanded CAG peak size, we used an internal standard to assign, on a per plate basis, a main CAG length to each sample. We used peak heights to quantify mosaicism from GeneMapper traces. To calculate the proportion of expanded products for each sample, we normalized the heights of the expanded peaks to the height of the main CAG peak, multiplied by the position of the peak. We applied a relative threshold of 0.03 of the main peak, excluding peaks falling below this threshold from analysis. We selected this threshold based on the additional peaks in fetal tissues that were low in intensity but clearly distinguishable from background by the software. Finally, we summed all peak values to generate an expansion index.

## Statistical analyses

We conducted all statistical analyses using R version 3.6.1 (*R Development Core Team, 2019*; https://www.R-project.org/), and we generated plots with the ggplot2 R package (*Wickham, 2009*) (ggplot2_3.3.0). We generated correlation plots using the corrplot R package (corrplot_0.84). The level of statistical significance was set at $p < 0.05$ for all tests.

## Descriptive statistics

Descriptive statistics were reported for the HD patients with demographics and disease characteristics (sex, age, somatic expansion index, and Unified HD Rating Scale total motor score [UHDRS-TMS]) determined at each visit that included blood sample collection. We defined AO as the onset of motor signs, as defined by the patient and their family, or first neurological exam at which they were considered symptomatic, whichever was earlier. Patients with a UHDRS-TMS greater than 5, which indicates motor signs of HD, were considered to have 'manifest' HD. We summarized the data as n (number of available values), mean ± SD, and range (minimum and maximum) for quantitative variables and frequency counts and percentages for categorical variables.

## Relationship between somatic CAG expansions and germline CAG repeat length

For samples collected from post-mortem brains (carrier fetuses and adult brain) or blood (two to three samples per patient), we studied the relationship between the CAG somatic expansions and the CAG repeat length by linear regression. We then determined the strength of association by the Pearson's correlation coefficient (r), the slope, and p-value of the regression line.

## Regression analysis of disease characteristics with CAG repeat and somatic expansion measures

### Expansion index (EI)

Prior to regression analysis, we transformed AO and AD values by the natural logarithm to better meet the linear model assumptions of normality and homoscedasticity (constant variance) of the residuals. Because we were able to collect blood samples at two or three time points for each patient in the longitudinal part of the study, we calculated corresponding EIs for each time point.

### Expansion rate (ER)

From these EIs, we were able to derive a rate of change in expansion over time (expansion rate or ER) in addition to the single time point measures. To investigate whether somatic instability itself evolves, i.e., whether the tendency to expand increases with age, both slope and intercept coefficients were extracted using linear regressions for each individual. We used the slope to calculate the

expansion rate of change (ER), while the intercept (EI-intercept) indicated a theoretical baseline value (age 0, i.e., at birth) for the expansion index.

### Expansion index at age at onset (EI-AO)

Even though the EI-intercept is too distant in time from the visits to be a realistic estimate of CAG instability at birth, we used the slope and the intercept for each patient to extrapolate a plausible (albeit theoretical) expansion index at AO (EI-AO).

In a first analysis, we performed linear regressions to model the values of log-AO and log-AD, respectively, as a function of CAG repeat length, EI-AO, and ER. We used the p-value of the slope and adjusted R squared ($R^2_{adj}$) values to determine all associations. Sex differences in AO and AD were also assessed using Wilcoxon rank-sum tests. Finally, we generated a correlation matrix plot summarizing all pairwise correlations between the variables from the longitudinal cohort.

Since the CAG repeat length is a well-established predictor of AO, we carried out the following analyses to understand whether combining information from the CAG repeat length and evolution of the somatic CAG instability could better characterize the disease onset.

### Determination of earlier-than-expected, as expected, or later-than-expected age at onset

Since AO, EI, and ER are all CAG length-dependent to a great extent, we sought a way to dissociate their contributions. To this end, we divided HD patients into three groups according to whether their motor symptom onset occurred earlier or later than the AO predicted by CAG repeat number [(CAG)n]. Following (*Swami et al., 2009*), we calculated the residuals from the linear regression, including log-AO as the dependent variable and (CAG)n as the independent variable, to evaluate the differences between the observed and predicted AO. We standardized residuals to have mean zero and unit variance and defined onset groups as 'earlier' for residual values less than −0.5, 'later' for residual values greater than 0.5, and 'as expected' otherwise. We then performed a Kruskal-Wallis test to compare the ER and EI-AO values among these groups.

### Relationship between somatic expansion and residual age at death

As a complementary analysis, similarly to the previous AO study, we used data from the 14 deceased patients in the longitudinal cohort, and data measured in the 14 postmortem brains (*Tables 2* and *3*). Based on the residual AD (i.e., AD after subtracting the effect of the CAGn using linear regression), we performed an association study with ER (blood samples) and EI at AD (postmortem samples) using linear regressions. Associations were reported with p-value of the slope and adjusted R squared ($R^2_{adj}$) values.

### Influence of disease status on EI and ER in blood samples

The cohort had a sufficient number of subjects in the premanifest and manifest stages at the first visit to study the influence of disease status on the residual EI and ER after using linear regression to subtract the contribution of CAGn. Since we could correlate EI with disease status only at the first visit (too many patients phenoconverted by the second visit), and because of the impossibility of clearly distinguishing the contributions of premanifest/manifest status, CAG repeat length, and age, this was an exploratory study prior to modeling using the complete expansion data with age and CAG repeat length. Comparisons of EI and ER with disease status were performed using Wilcoxon sum-rank tests.

### Distinguishing the determinants of somatic instability in blood samples: linear mixed-effects model

To investigate the longitudinal association of CAG repeat length and age with the somatic expansion in blood, we employed a linear mixed-effects model (LMM) including the variables age, CAG, and age × CAG interaction term as fixed effects, the subject identifier as a random effect to account for the within-subject correlation among visits ('random intercept only model'), and sex as a cofactor for adjustment. Prior to modeling, the somatic expansion values were transformed by natural logarithm to improve the model assumptions of linearity, normality, and constant variance of the residuals. LMM was fitted using restricted maximum-likelihood estimation (REML) from the function lmer

in the lme4 R package (*Bates et al., 2015a*) (lme4_1.1–21). For the retained model, we reported the coefficient estimates of fixed effects with standard errors and standardized regression coefficients (t values), and the standard deviation of random effects. T values were obtained by dividing each coefficient estimate by its standard error and used as a measure to represent the relative strength of association of each term with somatic expansion in blood. Significance of fixed effects (p-values adjusted for sex) was obtained with the lmerTest R package (lmerTest_3.1–1) using Satterthwaite's approximation for degrees of freedom. As age and CAG repeat length were mean-centered for modeling, the estimate for the model intercept can be interpreted as the level of somatic expansion for a virtual subject with average characteristics for all patients in the study (mean age and mean CAG repeat length). Curves for the age-trajectories of somatic expansion in blood (one trajectory per CAG value, *Figure 5*) were plotted from the fixed effects component of the model.

## Acknowledgements

We express our deepest gratitude to the patients and families participating in this study. Many thanks to Triplet Therapeutics for fruitful discussions and financial support. We thank Vicky Brandt for helpful discussions and editing the manuscript. We also thank research assistants Lynda Benammar, Marie Biet, and Rania Hilab for their engaged help and Ludmila Jornea and Philippe Martin Hardy for their laboratory skills. We received support from i-crin Neuroscience (NEUROLOP). This work was supported by grants from Agence Nationale pour la Recherche (Network of centers of excellence in neurodegeneration COEN, AD, SH); Fondation pour la Recherche Médicale (DEQ20170336752, SH).

## Additional information

### Competing interests

Alexandra Durr: serves on the advisory boards of Triplet Therapeutics. and holds partly a Patent B 06291873.5, "Anaplerotic therapy of HD and other polyglutamine diseases.". The other authors declare that no competing interests exist.

### Funding

| Funder | Grant reference number | Author |
| --- | --- | --- |
| Agence Nationale de la Recherche | ANR-16-COEN-0006-02 | Sandrine Humbert<br>Alexandra Durr |
| Fondation pour la Recherche Médicale | DEQ20170336752 | Sandrine Humbert |

The funders had no role in study design, data collection and interpretation, or the decision to submit the work for publication.

### Author contributions

Radhia Kacher, Conceptualization, Resources, Data curation, Formal analysis, Investigation, Methodology, Writing - original draft, Writing - review and editing; François-Xavier Lejeune, Software, Formal analysis, Methodology, Writing - original draft; Sandrine Noël, Resources, Software, Methodology; Cécile Cazeneuve, Conceptualization, Resources, Methodology; Alexis Brice, Validation, Investigation, Writing - original draft; Sandrine Humbert, Conceptualization, Supervision, Funding acquisition, Validation, Investigation, Writing - original draft, Writing - review and editing; Alexandra Durr, Conceptualization, Resources, Supervision, Funding acquisition, Validation, Investigation, Methodology, Writing - original draft, Project administration, Writing - review and editing

### Author ORCIDs

Radhia Kacher ⓘ https://orcid.org/0000-0003-4679-7361
Sandrine Humbert ⓘ https://orcid.org/0000-0002-9501-2658
Alexandra Durr ⓘ https://orcid.org/0000-0002-8921-7104

## Ethics

Human subjects: We obtained blood samples with written informed consent according to the French legislation (Approval from local ethics committees on 19/12/1990, 10/11/1992, followed by the Ethics committee Ile de France II on 30/9/2004 and 18/2/2010). Brain samples were collected as part of a program of 'Brain Donation for Research' (National Neuro-CEB Brain Bank, GIE Neuro-CEB BB-0033-00011).

## Decision letter and Author response

Decision letter https://doi.org/10.7554/eLife.64674.sa1
Author response https://doi.org/10.7554/eLife.64674.sa2

## Additional files

### Supplementary files

• Transparent reporting form

### Data availability

All patient data generated and analyzed are included in the manuscript and available in Tables 1, 2 and 3.

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
