## [Decision Letter]

**Acceptance summary:**

This is an important study evaluating blood and tissue somatic instability over long periods of time in several samples of patients with Huntington's disease (HD). The authors extensively revised the manuscript with more rigorous analyses and attention to messaging that better highlight contribution/relationship to the current concept that somatic instability is a major driver of HD disease.

**Decision letter after peer review:**

Thank you for submitting your article "Propensity for somatic expansions increases over lifetime in Huntington Disease" for consideration by *eLife*. Your article has been reviewed by 3 peer reviewers, one of whom is a member of our Board of Reviewing Editors, and the evaluation has been overseen by Huda Zoghbi as the Senior Editor. The following individual involved in review of your submission has agreed to reveal their identity: Vanessa C Wheeler (Reviewer #2).

The reviewers have discussed the reviews with one another and the Reviewing Editor has drafted this decision to help you prepare a revised submission.

We would like to draw your attention to changes in our policy on revisions we have made in response to COVID-19 (https://elifesciences.org/articles/57162). Specifically, when editors judge that a submitted work as a whole belongs in *eLife* but that some conclusions require a modest amount of additional new data, as they do with your paper, we are asking that the manuscript be revised to either limit claims to those supported by data in hand, or to explicitly state that the relevant conclusions require additional supporting data.

Summary:

This is an important study evaluating blood and tissue somatic instability over long periods of time in several samples of patients with Huntington's disease (HD). However, the manuscript needs to be revised considerably with more rigorous analyses and particular attention to messaging that better highlight contribution/relationship to the current concept that somatic instability is a major driver of HD disease.

Essential revisions:

1. As CAG length is a major modifier of instability index and instability rate, the associations of instability index and instability rate with AO, AD and residual AO (Pearson correlations in the text and Figures 4 and S2) are not particularly meaningful as these associations will capture CAG length-dependent effects to a great extent. This point should be discussed. To assess whether the degree of instability exhibited by an individual is associated with phenotypic outcomes, CAG repeat length and age need to be accounted for. i.e. based on regression models, a residual instability index or residual IR should be derived for association with phenotype. e.g. with residual AO or by testing significance in a statistical model for AO that also includes CAG length. The second approach was used in Ciosi et al. who found that instability was associated with AO and with TRACK progression score. It would also be important to test directly whether residual instability index and residual IR are different in the premanifest and manifest individuals, as a parallel approach to the statistical modeling that incorporates affected status.

2. Discussion of the association of manifest state on instability in the modeling is warranted. What is thought to underlie this? Presumably the ages of the manifest individuals differ significantly from the age of the premanifest individuals. Yet age, as a continuous variable in the model does not have a strong impact. Is it possible that manifest state as a categorical variable is capturing the impact of age? Or how rigorously can we rule out that age is independently accounted for? If Status is removed from the model, what is the impact of the age and age x CAG terms?

3. Given the longitudinal data and the phenoconversion of a large number of individuals over time can association with phenoconversion be performed within individuals?

4. Notably, there is no discussion of the data presented in the context of the Ciosi paper that has examined instability in blood from a large number of HD mutation carriers and correlated this with phenotypic measures. This should be part of the Discussion section.

---

## [Author Response]

Essential revisions:1. As CAG length is a major modifier of instability index and instability rate, the associations of instability index and instability rate with AO, AD and residual AO (Pearson correlations in the text and Figures 4 and S2) are not particularly meaningful as these associations will capture CAG length-dependent effects to a great extent. This point should be discussed. To assess whether the degree of instability exhibited by an individual is associated with phenotypic outcomes, CAG repeat length and age need to be accounted for. i.e. based on regression models, a residual instability index or residual IR should be derived for association with phenotype. e.g. with residual AO or by testing significance in a statistical model for AO that also includes CAG length. The second approach was used in Ciosi et al. who found that instability was associated with AO and with TRACK progression score.

These are excellent points: CAG length is indeed a major modifier of instability, and AO, AD, and residual AO will reflect this dependency. We found that creating a residual expansion index or expansion rate was not ideal, however, because it would take a much larger sample of hundreds or thousands of patients to achieve sufficient statistical power to discern correlation with AO, which is a rough estimate in HD.

In HD, AO is reasonably designated as the point at which motor symptoms declare themselves unequivocally; we followed this practice in our study, requiring a UHDRS-TMS score >5. However, motor symptoms may present only after years of cognitive or psychiatric deterioration. Thus, while motor-based AO may be appropriate for the patient who develops chorea first, it will not adequately account for the patient who has suffered HD-related cognitive or behavioral symptoms for ten years but has only just achieved a UHDRS-TMS of 5. Yet it would be even harder to pinpoint AO using cognitive or psychiatric abnormalities. AO, in other words, is doomed to fuzziness. Thought of in this way, it is rather remarkable that CAGn correlates with AO as strongly as it does—and more understandable that the correlation historically has ranged from ~41-72% or so. Unfortunately, we don't have full clinical information on all our patients such that we could use an overall disease severity score that incorporates multiple functional domains. As we revised the manuscript, we were careful to discuss AO, phenoconversion, and premanifest/manifest more carefully, so as not to over-simplify or obscure these realities for the broader readership of *eLife* who are not specialists in HD.

That said, to respond to this comment, we compared the residual AO, based on the CAGn, to the expansion rate. There was no correlation (Author response image 1). The same was true for residual age at death (AD) – another concept that is less straightforward than it appears, since the death may have been caused by suicide, accident, cancer, or another disease not related to HD. Another consideration is that our sample was weighted toward subjects with small repeat expansions, which have the greatest variance in AO.

**Author response image 1. sa2fig1:** 

Therefore, in the revised manuscript, we examined the potential association between expansion rate and AO using another method: we determined groups with earlier- or later-than-expected AO based on the regression between AO and CAGn (**Figure 3A, formerly 4A). We then correlated these groups with the expansion index estimated at AO and expansion rate (new Figure 4A,***left and right*, respectively). We based this approach on an analysis performed in Swami et al., *Hum Mol Genet* 2009. This method allowed us to accommodate the effect of CAG repeat length on AO and explore whether the variability in AO is associated with somatic expansion. Even though later than expected, groups have a tendency to have lower rates, it was not statistically significant, a similar longitudinal study on a larger cohort will be necessary to expand this study.

It would also be important to test directly whether residual instability index and residual IR are different in the premanifest and manifest individuals, as a parallel approach to the statistical modeling that incorporates affected status.

This was an excellent suggestion. The **new Figure 4B compares** the residual expansion index and expansion rate (note that we have changed our terminology) in premanifest and manifest individuals.

2. Discussion of the association of manifest state on instability in the modeling is warranted. What is thought to underlie this? Presumably the ages of the manifest individuals differ significantly from the age of the premanifest individuals. Yet age, as a continuous variable in the model does not have a strong impact. Is it possible that manifest state as a categorical variable is capturing the impact of age? Or how rigorously can we rule out that age is independently accounted for? If Status is removed from the model, what is the impact of the age and age x CAG terms?

The reviewer is correct that manifest state was likely capturing the influence of age, so we removed the premanifest/manifest parameter from the analysis (old Figure 5) and now use a linear mixed effect model to distinguish the contributions of age, (CAG)n, and age x (CAG)n (see Methods and **new Figure 5**). This approach finds that the tendency to somatic expansions depends first on CAG repeat number, then age, and finally on the interaction between age and CAG repeat number, and gives us a first glimpse of how somatic instability evolves in patients, using longitudinal sampling.

3. Given the longitudinal data and the phenoconversion of a large number of individuals over time can association with phenoconversion be performed within individuals?

We do find group differences between premanifest and manifest patients in terms of residual expansion index and expansion rate, but (as discussed in points 1 and 2 above) phenoconversion is too fuzzy to correlate with repeat instability on an individual subject basis with a sample population less than many hundreds.

4. Notably, there is no discussion of the data presented in the context of the Ciosi paper that has examined instability in blood from a large number of HD mutation carriers and correlated this with phenotypic measures. This should be part of the Discussion section.

The Ciosi paper was a landmark study of about 750 subjects that established, among other things, that somatic expansions are associated with worse clinical outcomes (earlier AO, higher motor score, higher progression score), and we credited it multiple times throughout the paper as its various findings were mentioned. We have completely revised the Discussion, however, in response to reviewer comments and better contextualize our results in light of both the Kaplan model and the Ciosi paper.